# Cocoa Bean Shell: A By-Product with High Potential for Nutritional and Biotechnological Applications

**DOI:** 10.3390/antiox12051028

**Published:** 2023-04-28

**Authors:** Marta Sánchez, Amanda Laca, Adriana Laca, Mario Díaz

**Affiliations:** Department of Chemical and Environmental Engineering, University of Oviedo, 33006 Oviedo, Spain; marta.ssotero@gmail.com (M.S.); lacaamanda@uniovi.es (A.L.); mariodiaz@uniovi.es (M.D.)

**Keywords:** cocoa bean shell, cocoa by-products, bioactive compounds, antioxidants, phenolic compounds, methylxanthines

## Abstract

Cocoa bean shell (CBS) is one of the main solid wastes derived from the chocolate industry. This residual biomass could be an interesting source of nutrients and bioactive compounds due to its high content in dietary fibres, polyphenols and methylxanthines. Specifically, CBS can be employed as a raw material for the recovery of, for example, antioxidants, antivirals and/or antimicrobials. Additionally, it can be used as a substrate to obtain biofuels (bioethanol or biomethane), as an additive in food processing, as an adsorbent and, even, as a corrosion-inhibiting agent. Together with the research on obtaining and characterising different compounds of interest from CBS, some works have focused on the employment of novel sustainable extraction methods and others on the possible use of the whole CBS or some derived products. This review provides insight into the different alternatives of CBS valorisation, including the most recent innovations, trends and challenges for the biotechnological application of this interesting and underused by-product.

## 1. Introduction

Large amounts of waste generated from food processing are discarded every year, which has become an increasing social, environmental and economic concern. In particular, 14% of food produced globally is lost from harvest up to retail, and 17% more is wasted at the retail and consumer levels [1,2]. The need to optimize the food chain in terms of sustainability and efficiency has emerged in recent years. In addition, the reduction and valorization of agri-food coproducts is one of the Sustainable Development Goals established by the Agenda 2030 [3,4,5]. In this context, there is a growing interest in wastes generated by the primary sector due to their high content of bioactive compounds, their low toxicity and great consumer acceptance [6].

Cocoa (*Theobroma cacao* L.) is one of the most important crops at the global level, with an annual worldwide production of around 5 million tons of dry cocoa beans. The main cocoa varieties are classified into four categories: *Criollo*, *Trinitario*, *Forastero* and *Nacional*, based on geographical origin and bean morphology. *Forastero* type is the variety most employed in commercial agricultural holdings (around 70% of total production in economic terms) [7,8]. In addition, the cocoa industry generates large amounts of residues; specifically, almost 20 tons of waste are produced for each ton of dry cocoa bean obtained. Admittedly, approximately 90% of total cocoa fruit weight is discarded as residue, whereas just 10% is marketed. This entails not only environmental but also economic impacts derived from waste management [8,9,10]. Figure 1 shows the cocoa steps processing and the by-products obtained.

The residual biomass derived from cocoa production and processing mainly consists of cocoa pod husk (CPH), cocoa mucilage (also known as the pulp) and cocoa bean shell (CBS). Cocoa pod husk is the residual pod material of the matured fruit once cocoa beans and placenta have been removed, and it represents between 70 and 80% of the whole fruit. It is composed of four layers: epicarp, mesocarp, sclerotic and endocarp, and it is an important source of bioactive compounds and fibrous material [12,13,14]. The mucilage is a white mass covering the cocoa beans that releases a turbid liquid (sweating) during post-harvest processing by means of microorganisms, such as yeasts, present in the cocoa fruit. The mucilage is rich in sugars and minerals, so it has great potential to be employed as a medium for the growth of interesting microorganisms at the industrial level [15,16]. CBS is the outer part surrounding the cocoa bean and represents between 10 and 20% of the total cocoa bean weight. This by-product contains carbohydrates, dietary fibres, fats, phenolic compounds, antioxidants, and vitamins. CBS is commonly considered to be a residue, being discarded or employed in low-value applications (animal feed or fertilizer). CBS is the main by-product of chocolate industries, separated from the cotyledons during the pre-roasting process or after the roasting process [17]. In recent years, there has been a growing interest in obtaining value-added products from CBS, with applications in the food, cosmetic, pharmaceutical or biofuel sectors not only due to its specific characteristics but also because its exploitation can be economically attractive [18,19,20,21]. Previous reviews related to CBS have been specifically focused on particular aspects of this by-product, namely, its role in human health [7], its use as a source of bioactive compounds [11,16] or its utilization in food products. The present work addresses the potential alternatives to valorize CBS with the aim of providing an update on this interesting topic from a global perspective, i.e., including from its use as animal foodstuff to employing it as a substrate for bioplastic production.

## 2. Chemical and Nutritional Composition of CBS

CBS is the fibrous external tegument and constitutes about 10–20% of the total cocoa bean weight [10,22,23]. CBS is recovered during the cocoa bean roasting process at the chocolate factory, i.e., for every kg of chocolate produced, almost 100 g of CBS is obtained [24,25,26]. It is estimated that each year, between 700 and 900 thousand tons of CBS are produced worldwide, approximately 300 of which are generated in Europe. The accumulation of this by-product represents an important disposal problem in terms of economic and environmental issues, which is further accentuated by new legal regulations [26].

Although CBS is generally considered and treated as a residue, it has been reported that its nutritional composition does not differ too much from that of cocoa beans. CBS nutritional composition has been deeply investigated in different research works with the aim of extracting carbohydrates, dietary fibres, phenolic compounds (mainly epicatechin and catechin) and methylxanthines, such as theobromine. Table 1 summarizes the composition of CBS according to literature data, which varies depending on the origin and the processing of cocoa fruit.

It has been reported that the moisture content of CBS, which is determined by the roasting process, normally ranges from 4 to 8 g/100 g, values considered acceptable for CBS storage. However, in some cases, moisture levels can reach up to 13 g/100 g [29]. In this sense, CBS has been described as a considerably hygroscopic material; therefore, storage at moisture levels over 15% could lead to mould growth [27]. According to the literature, the ash content is in the range of 6 to 9 g/100 g of CBS, which is a parameter also influenced by the roasting process and is relatively similar among varieties. CBS contains a higher ash content than other fruit residues, such as apple pomace (0.5 g/100 g) and orange peel (2.6 g/100 g) [52,53]. Osundahunsi et al., 2007 [54] reported that sodium and potassium are the main components of CBS ash.

The fat content is quite low in CBS compared to cocoa beans, which accounts for almost 50% (dry weight). It has been reported that the content of lipids in CBS can be reduced by 40% after the roasting process [29]. The chemical and physical characteristics of CBS lipids are similar to cocoa butter fat, except for the acidity levels, which are higher in CBS (9% oleic acid) than in cocoa butter (1.7% oleic acid) due to the hydrolysis of triacylglycerols [49]. The saponifiable fraction of CBS is mainly formed by oleic, capric, palmitic and stearic fatty acids, whereas the unsaponifiable fraction is composed of phytosterols, specifically stigmasterol [20]. From a health perspective, it has been demonstrated that the consumption of cocoa increases high-density cholesterol and reduces low-density cholesterol because of its fatty acid content [55].

The carbohydrate content in CBS varies between 13.2 and 70.3 g/100 g (Table 1), mainly constituting glucose (17%), galactose (3%), mannose (3%), xylose (1.2%) and arabinose (1.7%) [10,30]. Carbohydrates are one of the most important macronutrients in cocoa beans since these compounds are responsible for the flavour during the fermentation and roasting processes. Additionally, it has been demonstrated that roasted CBS contains more carbohydrates than unroasted CBS, which is associated with the transfer of sugars toward the outer shell during roasting [29]. Furthermore, it has been reported that some CBS carbohydrates, such as feruloylated oligosaccharides, could improve the intestinal microbiota due to their prebiotic characteristics [53]. With respect to the protein content, during fermentation, an increase of up to 30% can be achieved in CBS. However, roasting processes result in a reduction in the protein content of CBS. According to the literature, proteins constitute between 18.2 and 27.4% of CBS. Furthermore, it is known that just approximately 1% of the total protein in CBS remains in a free condition, while 90% is firmly bound to oxidized polyphenols found in the shell [49,56].

CBS has been reported to be an important source of dietary fibre (13.8–65.6 g/100 g) with a small fat content (2–7 g/100 g). The amount of dietary fibre in CBS also depends on the roasting process. Certainly, it has been proved that, in roasted cocoa shells, Maillard reaction compounds and tannin or protein complexes are responsible for an increment in fibre content [57]. Insoluble dietary fibre accounts for 80% of the total dietary fibre of CBS extracts and almost half of the total dry weight. According to the literature, the main components of this CBS insoluble fibre are uronic acid and glucose, followed by galactose, xylose, mannose and arabinose to a lesser extent. This composition indicates that the cell wall mainly consists of cellulose with lower amounts of hemicellulose and pectin [17]. It has been reported that cocoa fibre has physicochemical and antioxidant properties, which make it an interesting alternative for the preparation of low-calorie and high-fibre cocoa products [57].

Along with dietary fibre, CBS polyphenols are the most studied and interesting compounds since they are responsible for biofunctional properties attributed to this by-product, such as its antidiabetic, anticarcinogenic and anti-inflammatory effects [58]. Flavonoids, one of the most remarkable groups of phenolic compounds, have been widely investigated due to their extensive bioactive properties, including cardio-protective, anti-oxidation and anti-inflammatory activities [59]. The reviewed literature describes that the total phenolic content (TPC), total flavonoid content (TFC) and total tannin content (TTC) in CBS range between 22 and 100 mg GAE/g CBS, 1.6 and 44 mg CE/g CBS and 2.3 and 25.3 CE/g CBS, respectively. The number of phenolic compounds in CBS extracts varies depending on several factors, such as origin, variety, extraction procedure (technologies, conditions, solvents…) and the presence of other bioactive compounds [40,60,61]. Flavanols have been identified as the major class of flavonoids in CBS, mainly including catechin (0.8–5.7 mg/g CBS), epicatechin (0.6–30 mg/g CBS) and procyanidin B2 (0.2–1.4 mg/g CBS). These polyphenols are not essential for short-term well-being, but it has been suggested that a long-term intake of them could contribute to health benefits, such as those related to cognitive function [62]. Other flavonoids also detected in CBS include kaempferol, quercetin and anthocyanins [26,40,58].

Theobromine and caffeine are the main methylxanthines, also known as alkaloids, found in CBS in quantities that fluctuate from 0.6 to 2.7 mg/g of CBS and 0.1 to 0.1–1.1 mg/g of CBS, respectively. Both alkaloids are related to physiological effects on the cardiovascular, gastrointestinal, respiratory, renal and central nervous systems in addition to their anticarcinogen and diuretic properties [20,61]. However, it is well known that excessive consumption of methylxanthines, particularly caffeine, is related to tachycardia, kidney dysfunction and other disorders. In addition to theobromine and caffeine, theophylline is another methylxanthine detected in CBS extracts, but at lower concentrations (0.1–0.3 mg/g CBS) or even at trace levels [51,60]. Methylxanthines, along with polyphenols, contribute to the characteristic bitter taste of cocoa and its derivatives, such as chocolate [63].

CBS can also contain some toxic compounds and/or antinutrients, such as biogenic amines, trypsin inhibitors and phytic acid. However, they are at concentrations so low (for example, the amount of phytic acid is below 0.6 g/100 g CBS) that this does not affect the use of CBS as a potential source of nutrition [53,64].

## 3. CBS Valorization

Recently, CBS valorization has been investigated to find novel applications for this by-product, including its use in food formulation, the obtention of biofuels, the extraction of bioactive compounds and employment as an adsorbent. Figure 2 shows an overview of different potential applications of CBS.

### 3.1. Bioactive Compounds

Research on bioactive compounds has received great interest due to the essential role of these substances in reducing the risk of several diseases. Specifically, it has been reported that the intake of specific bioactive compounds is related to reducing the risk of cancer, degenerative diseases or cardiovascular disorders [65].

CBS extracts contain bioactive fractions that include carbohydrates, dietary fibre, proteins, lipids and secondary metabolites. For example, it has been suggested that cocoa cell wall fibre possesses health-promoting properties, such as decreasing total cholesterol, insulin and triglyceride levels [66]. Furthermore, CBS contains bioactive peptide fractions, specifically albumin (12%) and vicilin (4%), with antioxidant, anti-obesogenic and anti-diabetic attributes [67]. In addition, phytosterols, such as stigmasterol or campsterol, identified in CBS, show anti-inflammatory, anti-tumoral, anti-bacterial, and anti-fungal activities and also reduce blood cholesterol levels [53]. For these reasons, the use of this by-product as a source of bioactive compounds has attracted strong interest in recent years.

Among the bioactive compounds present in CBS, antioxidant, antimicrobial and antiviral compounds are the most interesting groups; therefore, they are commented on in detail in the sections below.

#### 3.1.1. Antioxidant Compounds

Antioxidant compounds are well-known for their crucial role in providing protection against oxidative damage, which is closely linked to the development of human chronic and degenerative diseases [2,68]. Their mechanism of action consists of the scavenging of reactive oxygen species of cellular metabolism and preventing damage to biologically relevant molecules, such as DNA, proteins and membrane lipids and cells [69]. For example, Ávila-Gálvez et al., 2019 [70] studied the metabolic profile of different polyphenols in vivo in normal and tumoral breast tissues from patients that consumed capsules with CBS extracts. These authors determined that these extracts exhibited anticancer and antiproliferative activities compared to control patients. Moreover, Aranaz et al., 2019 [44] investigated the favorable effects of cocoa extracts (containing epicatechin, catechin and procyanidin B2) on lipid metabolism and disorders such as obesity and metabolic syndrome. The former authors also reported beneficial results in terms of insulin resistance, liver steatosis and glucose intolerance in study groups supplemented with cocoa extract in contrast to non-supplemented study groups. Phenolic compounds are the most relevant antioxidants present in CBS. As a result of processing steps such as fermentation and roasting, phenolic compounds migrate from cocoa seed cotyledons to the CBS, generating polyphenol-enriched cocoa shells. Additionally, Rossin et al., 2021 [71] reported the capacity of CBS-enriched ice cream to protect the intestinal cell layer from the damage associated with inflammatory and oxidative reactions.

As previously mentioned, phenolic compounds are the main group of secondary metabolites in CBS, playing an essential role in preventing several diseases due to their antioxidant properties. The main polyphenols found in CBS are flavonoids, in particular flavanols, which include epicatechin, catechin and procyanidins [49]. Cocoa flavanols act as chemopreventive agents against neurodegenerative diseases, cancer, heart disorders and ageing. Among the mechanisms involved in these preventive effects, the stimulation of tumor suppressor genes and the activation of the insulin pathway are remarkable [72,73,74]. It has been confirmed that CBS flavanols, specifically procyanidins, are implicated in the regulation of the cancer-signal transduction pathways associated with differentiation, apoptosis, inflammation, cell proliferation, angiogenesis, or metastasis [72]. In addition, some studies have revealed that cocoa polyphenols can prevent the progression of prostate, colon, liver, breast or lung cancer, among others [75,76,77].

The amount and type of phenolic compounds in CBS are usually linked to the origin and variety of cocoa. For example, Barbosa-Pereira et al., 2021 [78] studied the chemical and nutritional composition of ten different samples of CBS from different areas of Venezuela and two varieties (*Criollo* and *Trinitario*) and obtained an amount of TPC and TFC in the range of 5.8 to 7.5 mg GAE per g of CBS and 1.8 to 3.6 mg CE per g of CBS, respectively. In addition, Martínez et al., 2012 [32] evaluated the antioxidant properties of CBS from two different locations in Ecuador (Cone and Taura) and obtained 2.5 µM Trolox equivalent (TE)/g of CBS from Cone, almost half the amount obtained from Taura CBS (4.5 µM TE/g CBS).

Moreover, the extraction methodology employed is also determinant in relation to the number of antioxidant compounds recovered from CBS. In this sense, traditional extraction techniques using organic solvents are the most employed for CBS. For instance, Nsor et al., 2012 [39] evaluated the TPC and antimicrobial activity of CBS extracts obtained with different solvents (water, ethanol, methanol and acetone). They reported that the highest number of phenolic compounds (41.8 mg GAE/g) were observed when acetone was employed, whereas when using the most polar solvent (water), they obtained the minimum TPC value (17.2 mg GAE/g). For ethanolic and methanolic extracts, similar values were achieved (23.3 and 25.1 mg GAE/g, respectively). However, Rossin et al., 2019 [79] also tested different extraction solvents (water, methanol/water (50:50 (*v*/*v*)), methanol and acetone) with CBS powder at 25 °C stirring for 1 h, and obtained the highest concentration of methylxanthines and TPC in the aqueous extract (8 mg/mL and 546.7 mg GAE/l, respectively). In a similar way, Manzano et al., 2017 [80] obtained a maximum amount of TPC of 7 mg GAE/g CBS employing water as solvent by stirring for 5 min. On the other hand, Papillo et al., 2019 [48] investigated the extraction of phenolic compounds from CBS through stirring and an ultrasound treatment for 30 min. They obtained in ethanolic extracts approximately twice as much antioxidant activity (215 mg TE/CBS) than that obtained in water extracts. In addition, Hernández-Hernández et al., 2019 [61] compared the effectiveness of five different methods of polyphenol extraction from CBS. Methanol (80% (*v*/*v*)) and water, with methanol (80% (*v*/*v*)) at pH 3 and water at pH 3, were employed as solvents when stirring the samples for 1 h at 70 °C. In addition, CBS was treated with ethanol/acetone (70:30 (*v*/*v*)) when stirring the samples for 2 h at room temperature. Acidified methanol and ethanol/acetone were the most effective solvents for the extraction of TPC from CBS (11 and 13 mg GAE/g CBS, respectively).

Conventional extraction methods conducted to recover phenolic compounds from plant-based matrices are normally time-consuming and require a high consumption of solvents and energy. In addition, they can lead to the oxidation and denaturation of polyphenols [2,28,65,81,82]. Therefore, in recent years, environmentally friendly extraction methods, which are capable of recovering high amounts of bioactive compounds with short extraction times and low solvent consumption, are gaining interest [17]. Specifically, green technologies, such as ultrasonic-assisted extraction (UAE), pulsed electric fields (PEF), microwave-assisted extraction (MAE) or supercritical fluid extraction (SFE), have been employed to obtain different compounds of interest from CBS. For example, Grillo et al., 2019 [30] used ultrasonic-assisted extraction to obtain CBS extracts with an amount of 51 mg GAE/g and 8 mg/g of TPC and methylxanthines, respectively. Agudelo et al., 2021 [83] subjected CBS to the UAE methodology and recovered a maximum amount of 8.3 mg/g of theobromine, 0.17 mg/g of catechin and 3.8 mg/g of epicatechin. Jokić et al., 2018 [60] employed subcritical water extraction (SWE), obtaining a maximum of 130 mg GAE/g CBS, and reported the detection of alkaloids (theobromine, theophylline and caffeine), catechin, epicatechin and chlorogenic acid in the resulting broths. A recent study using PEF for the extraction of CBS antioxidants reported an increase of 20% in the phenolic compounds and antioxidant activity compared to conventional extraction methods [39]. Carpentieri et al., 2022 [84] also employed PEF to obtain CBS ethanolic extracts with a maximum concentration of theobromine of 2 mg/g CBS. In addition, Mellinas et al., 2020 [83] obtained extracts from CBS by means of MAE (5 min, 97 °C) with an amount of TPC of 35.5 mg GAE/g and an antioxidant activity of 33.6 mg TE/g, values 50–62% higher than those obtained through conventional heating. Jensch et al., 2022 [85] recovered a maximum of 5 mg/g of catechin and epicatechin from CBS submitted to pressurized hot water extraction (PHWE) at 140 °C. Table 2 summarizes different phenolic compounds obtained from CBS as well as the extraction techniques employed for this purpose.

#### 3.1.2. Antimicrobial Compounds

Antimicrobial compounds could be defined as natural substances with the capacity to inhibit the development of microorganisms by means of altering their metabolism [2]. Recently, a rising interest in natural compounds that inhibit the growth of microorganisms involved in food deterioration and foodborne illness has arisen. Moreover, the increasing resistance of some bacteria to drugs has led to the urgent need to obtain new compounds that can replace conventional antibiotics [86,87]. In this context, in recent years, the potential antimicrobial activity of plant-derived substances in terms of antimicrobial, antioxidant and anti-inflammatory functions has been studied. The antimicrobial capacity of phenolic compounds is based on the disruption of the cell wall, which increases membrane permeability, leading to the release of cell constituents.

It has been published that CBS extracts are an interesting source of antimicrobial compounds. Nsor-Atindana et al., 2012 [39] evaluated the antibacterial activity of CBS extracts (employing water, acetone, ethanol and methanol as solvents) against *Escherichia coli*, *Salmonella* sp., *Staphylococcus aureus* and *Bacillus cereus*. Minimal inhibitory concentrations (MICs) of the extracts ranged between 0.5 and 3.7 mg dry extract/mL. The results showed that all of the extracts exhibited antimicrobial power against all of the strains tested, with the acetone extract presenting the highest inhibition effect. The aqueous extract exhibited a lower antimicrobial effect, whereas the ethanolic and methanolic extracts achieved similar inhibition values. These results are in accordance with those obtained by Kayaputri et al., 2020 [88], who reported the inhibitory effects of CBS extracts against *Salmonella* sp., *S*. *aureus* and *E.coli*, allowing for a reduction in the total bacteria count of 6%, 7% and 20% for *E. coli, S. aureus* and *Salmonella* sp., respectively. Tamrin et al., 2020 [89] reported similar antibacterial activities related to CBS extracts (aqueous, ethanolic, methanolic and acetone) against *E. coli*, *B. cereus* and *S. aureus* (approximately 7 mm inhibition zone). Moreover, Rojo-Poveda et al., 2021 [90] tested the antimicrobial capacity of CBS extracts at different concentrations (0.004 to 0.5 mg of dry extract per ml) in eight bacterial and fungal strains. These authors found that inhibition was only detected at the maximum concentration (0.5 mg/mL) against *E. coli*, *Pseudomonas aeruginosa*, *Saccharomyces cerevisiae*, *Candida albicans*, methicillin-resistant *S. aureus* (MRSA) and methicillin-sensitive *S. aureus* (MSSA). Only *Streptococcus mutans* showed inhibition at pharmacologically interesting concentrations (below 0.5 mg/mL). In the case of S. cerevisiae, not only did it not show any inhibition, but an increase in growth was observed in this specie when CBS concentrations increased, which is in accordance with the natural presence of this yeast in the cocoa bean fermentation process.

Additionally, some works have revealed the antibacterial properties of CBS against *S. mutans*, a bacterial strain responsible for dental caries. Babu et al., 2011 [91] compared the potential antimicrobial activity of chlorohexidine (CHX) and CBS-based mouth rinse against *S. mutants*. They reported similar results for both rinses with respect to *S. mutans* counts in saliva at all of the intervals analyzed. Thus, these authors concluded that CBS extract could be an alternative to CHX mouth rinse, avoiding the side effects of CHX, such as a burning sensation or unpleasant taste. The anti-caries activity of polyphenols obtained from plants is based on the inhibition of S. mutans and the inhibition of glucosyltransferases (GTFs), an enzyme associated with the formation of dental plaque glucans on the tooth surface [92]. Percival et al., 2006 [93] observed the anticariogenic effect of CBS acetone extracts on the dental biofilms of S. mutans and *Steptococcus sanguinis*. In addition, Osawa et al., 2001 [94] described the antibacterial effect of ethanolic extracts obtained from CBS on S. mutans and *Streptococcus sobrinus* and their anti-GTF activity. Matsumoto et al., 2004 [95] analyzed the inhibitory effects of CBS extracts on dental plaque formation in vitro and in vivo studies. The results indicated that CBS ethanolic extracts could reduce the adherence of S. mutans to saliva-coated hydroxyapatite, an artificial dental plaque, and its amount in plaque in vitro. In addition, when CBS extracts were used as a mouth rinse in an in vivo study, inhibited plaque deposition on the tooth surface was observed.

#### 3.1.3. Antiviral Compounds

The lignin–carbohydrate complexes (LCCs) formed in the cell walls of CBS have been reported to be related to antiviral effects. For example, it has been found that LCCs inhibit the cytopathic effect of immunodeficiency virus (HIV) in cell culture [96]. Furthermore, Sakagami and Matsuta (2013) [97] confirmed that LCCs from CBS have higher anti-HIV activity (selective index (SI) = 30–10,000) than flavonoids (SI = 1), tannins (SI = 1–10) and natural lignin from other plant-based matrices (SI = 10–100). Unten et al. observed that when CBS extracts were added to the cells at the HIV virus adsorption step, maximum antiviral activity was achieved. In addition, cocoa procyanidins have also exhibited antiviral properties against influenza A virus and herpes simplex virus (HSV), with a selective index of approximately 155. The mechanism of action is believed to consist of avoiding the entry of the virus into the host cell, a critical step in virus infection [98].

### 3.2. Biomaterials

Preservation and food lifetime extension are some of the main priorities in terms of reducing food waste; thus, plastic food packaging seems to be essential for the preservation of comestibles. Synthetic plastics, such as polyamides, polyolefins and polyesters, are usually employed in food packaging [99,100]. However, environmental pollution risks are commonly associated with these petroleum-derived materials. In this context, natural polymers from renewable resources have emerged as a sustainable alternative due to their high biocompatibility, low environmental effects and rapid degradation [101]. One of the bioplastics most used in packaging is poly-(lactic acid) (PLA), an aliphatic biodegradable and thermoplastic polymer derived from renewable sources [102,103].

In recent years, CBS has been investigated in terms of food packaging applications due to its flexural and tensile mechanical properties, reduced density and low cost. For example, Papadopoulou et al., 2019 [104] developed biocomposites from a mixture of PLA and CBS powder. The results showed an improvement in the physical properties of the resulting composites, as well as in their biodegradability in aqueous media because of the inclusion of CBS. The use of this by-product also conferred antioxidant characteristics to the PLA/CBS composites. Puglia et al., 2016 [105] proposed the introduction of CBS into a biodegradable matrix as polycaprolactone (PCL), which successfully enhanced the material properties in terms of a strong interface and the rigidity of the polymer. In addition, García-Brand et al., 2021 [106] evaluated the use of CBS-based composites in biomaterial formulation. These authors reported that PLA/CBS composites are highly biocompatible in terms of hemocompatibility, cytotoxicity and antioxidant properties, which leads to the possible use of CBS in bioactive material production for application in the biomedical field.

Other biodegradable polymers used as an alternative to synthetic polypropylene are polyhydroxyalkanoates (PHAs) [107,108]. They are produced through bacterial fermentation when the external conditions are unstable. Poly(3-hydroxybutyrate) (PHB), which is the most common PHA, is usually produced through the action of microorganisms such as *Bacillus*, *Pseudomonas* and *Cupriavidus* when the external supply is scarce as an energy reserve [109,110]. The main limitation regarding the industrial production of PHB is related to carbon source costs. Thus, it is essential to find novel and sustainable carbon source alternatives, such as lignocellulosic residual biomass. In this context, Sánchez et al., 2023 [10] studied, for the first time, the use of CBS hydrothermally treated as feedstock to obtain PHB employing *Bacillus firmus.* Two different broths, non-centrifuged (with CBS solids) and centrifuged, were employed as fermentation media. A maximum PHB concentration of 20 g/L was achieved in the non-centrifuged medium, whereas very low PHB production (0.6 g/L) was obtained in the centrifuged medium. These results highlight the fundamental role of CBS solids in microorganism metabolism, being essential for PHB production.

### 3.3. Adsorbent

Emerging pollutants, including heavy metal ions, industrial dyes, microplastics and pharmaceutical compounds, imply environmental concerns, which are mainly related to water pollution. Several technologies have been employed to remove emerging pollutants from wastewater, such as ozonation, chlorination, biodegradation and adsorption [111,112,113,114,115]. Among them, adsorption with activated carbons has been extensively applied due to several advantages, including ease of operation, lower operating costs compared to other methods and high removal efficiency [116,117]. Nevertheless, researchers continue to explore alternatives to commercial activated carbons in order to reduce the material cost associated with treatment processes [118]. In this context, agri-food wastes, such as CBS, have been widely investigated as novel adsorbents [119,120].

Al-Yousef et al., 2021 [118] evaluated the application of CBS as an adsorbent to eliminate ibuprofen (IBP) from water, and the results showed that the adsorption capacity of CBS in removing IBP was significantly higher than other materials, such as biochar, zeolites or even activated carbon. Moreover, Rodriguez-Arellano et al., 2021 [121] assessed the properties of dried CBS powder as an adsorbent for Congo red dye removal in water, achieving a maximum percentage of adsorption of 96% under optimal conditions (40 mg/L of dye concentration, pH 3, 0.11 g of adsorbent/L and 36 h). These authors indicated that the dye attached to the CBS surface, particularly, it was associated with the OH group of phenolic compounds. Additionally, Ribas et al., 2014 [122], who employed activated carbon obtained from CBS as an adsorbent for a commercial textile dye, reported removal values of 95% in aqueous media. El Achaby et al., 2018 [123] extracted cellulose microfibrils (CMFs) from CBS and employed them as adsorbents for the removal of dyes, such as methylene blue, from wastewater, achieving a maximum adsorption capacity of 382 mg/g.

Additionally, Kalaivani et al., 2015 [124] found that CBS activated carbon obtained at different temperatures presented interesting adsorbent properties for the elimination of Ni (II) ions from an aqueous medium. These authors remarked that carbon prepared at a higher temperature (350 °C) showed an adsorbent efficiency in terms of Ni (II) ion removal that was 62% higher with respect to the carbon obtained at 30 °C due to its small particle size and high surface area as a result of the heat treatment.

### 3.4. Biofuels

Environmental and economic concerns deriving from fossil fuel depletion have led to the development of new energy alternatives, namely, biofuels, leading to a transition away from fossil fuel use [125]. Lignocellulosic residues are interesting substrates for obtaining biofuels due to their high carbohydrate content [2]. Awolu and Oyeyemi (2015) [126] investigated the use of CBS hydrolyzed under different time and temperature conditions to obtain ethanol by employing S. *cerevisiae*. In addition, they optimized the fermentation process using three different variables: pH, yeast concentration and fermentation time. The results showed that using a neutral pH had a notable positive effect on ethanol production (9%) within 144 h when using a microbial load of 0.05 mg/L. Shet et al., 2018 [127] also studied the optimization of bioethanol production using CBS subjected to acid hydrolysis with 10% H_2_SO_4_ and obtained a maximum of 3.2 g/L of bioethanol under optimal conditions (6% solid/liquid ratio (*v*/*v*), acid treatment for 8 min, (2% (*v*/*v*) inoculum of *Pichia stipitis* and 72 h of fermentation). In addition, Mancini et al., 2016 [128] studied the use of CBS pretreated with N-methylmorpholine-N-oxide (NMMO) to obtain methane and reported an increase of 14% in terms of biomethane yield in the pretreated CBS (226 mL CH_4/_g VS) compared to the untreated samples (199 mL CH_4_/g VS).

Thompson and Rough (2021) [129] evaluated the use of CBS as an alternative bioenergy source when used as a pellet. They evaluated different parameters, such as CBS particle size, compaction speed, compaction stress and binder addition (water, bentonite clay and magnesium stearate), in order to obtain a quality pellet. The results showed that pellets with bentonite clay as a binder achieved a calorific value (16 MJ/kg) above the value required by ISO 17225-6 standards (14.5 MJ/kg).

### 3.5. Corrosion Inhibitor

One of the main aspects that the carbon steel industry focuses on is low corrosion resistance. Therefore, new processes need to be implemented in order to provide better protection against corrosion, and the use of inhibitors is one of the most suitable options [130,131]. Corrosion inhibitors could be defined as substances that, when added in low concentrations to a corrosive environment, are able to minimize the reaction between the metal and the medium, adsorbing organic molecules and ions on the metal’s surface and generating a protective layer [132]. However, many synthetic inhibitors employed in industry are toxic and expensive. For these reasons, recently, research on alternative natural inhibitors, which are environmentally friendly and low-cost, has notably increased.

It has been reported that the inhibitory properties of “green” inhibitors are linked to the presence of bioactive compounds, such as flavonoids, tannins, or alkaloids [133,134]. Consequently, CBS can be an excellent alternative to synthetic inhibitors because of its composition, which is rich in phenolic and bioactive compounds. De Carvalho et al., 2021 [135] tested different concentrations of CBS powder and hydroalcoholic CBS extract (80% (*v*/*v*)) as corrosion inhibitors for SAE 1008 carbon steel in a sodium chloride solution. The results showed that the corrosion rates of the steel decreased when the inhibitor concentration increased. In addition, the authors reported that for all of the concentrations tested, CBS powder did not show efficient corrosion, while hydroalcoholic extract presented a good performance at the lowest concentration (0.4 g/L). The same authors [136] evaluated the corrosion-inhibitory properties of CBS powder with the same steel but, in this case, in an acidic medium (HCl 0.5 mol/L). An excellent corrosion inhibition efficiency (96%) was achieved at the lowest concentration of inhibitor tested (0.4 g/L), revealing the potential of CBS as an efficient corrosion inhibitor for carbon steel.

### 3.6. Food Ingredient

Due to its nutritional and nutraceutical properties provided by its high content of dietary fibre and polyphenols, CBS has been largely proposed as an ideal ingredient or additive in food production.

For example, soluble dietary fibre from CBS has been employed to replace 50–70% of vegetable oil in muffin production. The results showed decreased hardening during storage, higher moisture, good texture and pleasant colour and acceptance of the muffins containing CBS [137,138]. In addition, the presence of volatile organic compounds (10–20%) in CBS composition, some of them related to chocolate-specific flavour, makes CBS an excellent and low-cost material to obtain a cocoa aroma. In this context, CBS has been widely used in bakery products, such as wheat bread, biscuits or cakes, providing a softening effect and increasing their fibre content and antioxidant properties [39,139]. Rinaldi et al., 2020 [140] studied three ranges of particle size of CBS for the development of improved gluten-free bread formulations and concluded that this by-product could be a promising functional ingredient. Additionally, Kãrklina et al., 2012 [141] evaluated the application of CBS powder as a substitute for wheat flour in butter biscuit production, increasing their nutritional value (5% proteins, 46% dietary fibre and 14% fats with respect to the control biscuits). Moreover, Nogueira et al., 2022 [142] proposed the elaboration of chocolate cake with different percentages of CBS powder (25–100%) replacing wheat flour. Cakes obtained employing 75% CBS powder presented satisfactory sensorial and nutritional properties (100 g TE/g CBS, 92 mg GAE/100 g CBS and 7.8 mg anthocyanins/100 g CBS). Bariŝić et al., 2021 [143] employed untreated and treated CBS with high-voltage electrical discharge (HVED) at different proportions (0–15%) to produce enriched chocolate. They reported that milk chocolate with 5% and dark chocolate with 15% of CBS, both untreated and treated, obtained a positive acceptance of consumers mainly due to the softness and darkness of the final products.

Several vegetable grains, such as wheat grains, contain high numbers of polyphenols [144]. However, these compounds are mainly confined in the cell wall, being little accessible for use by the human body during the digestive process. In this context, enzymatic treatments of these plant-based matrices enhance the bioavailability of phenolic compounds. Among processes for enzyme production, solid-state fermentation (SSF) has been receiving attention due to its sustainable characteristics, namely, low cost, use of agri-food wastes and high process yield. It has been reported that the use of CBS as a substrate to obtain enzymes through SSF using *Aspergillus awamori* as a microorganism and the successful application of the obtained enzymes in bread [145].

Encapsulation approaches for CBS extracts have been investigated in order to maintain the stability of its antioxidant compounds and, thus, obtain polyphenol-enriched food products with antioxidant properties. Papillo et al., 2019 [48] proposed spray-drying as a technique for CBS extract encapsulation employing maltodextrins as stabilizing agents for its use in bakery products. The biscuits containing encapsulated CBS presented a high content in terms of phenolic compounds that remained after the baking process and storage (up to 90 days). Similarly, Grassia et al., 2021 [146] obtained microcapsules of enriched CBS extracts (16 mg GAE/g for TPC) using a spray dryer and incorporated them for the preparation of chocolate bars. In addition, Altin et al., 2018 [147] encapsulated CBS extract into chitosan-coated liposomes, which ensured the stability of the content of TPC, TFC and antioxidant activity in drinking yogurt preparation during storage.

CBS has also been proposed as a raw material to obtain beverages such as homemade functional beverages, carbonated soft drinks or dairy drinks. For example, Quijano-Aviles et al., 2016 [148] optimized the use of CBS, coffee silverskin and orange peel as ingredients to formulate a dairy drink. They reported an optimal formulation containing 74% CBS, 25% coffee silverskin and 2% orange peel that yielded a final product with great antioxidant activity (82% inhibition) and maintained acceptable sensorial attributes (taste, colour and appearance). Additionally, Rojo-Poveda et al., 2019 [27] investigated different parameters (CBS particle size and extraction methods) for the production of a functional beverage from CBS. The results showed that the smallest grinding degree (250–500 µm particle size), along with percolation techniques, allowed the obtainment of beverages with the highest functional character, i.e., with the maximum TPC content (1803 mg GAE/l) and antidiabetic properties (52% α-glucosidase inhibition).

Siow et al., 2022 [149] studied the use of CBS from different kinds of cocoa beans (*Criollo* and *Trinitario*) from Malaysia, Vietnam and Venezuela roasted at different temperatures (100, 120 and 150 °C) to obtain cocoa tea. They reported that the antioxidant properties decreased as the roasting temperature increased, also affecting the flavour of the final tea product. Malaysia cocoa tea showed the highest amount of TPC and antioxidant activity (19.4 mg GAE/g and 23.1 mg ascorbic acid (AA)/g, respectively), whereas the Venezuelan cocoa tea presented the maximum concentration of methylxanthines (165 µg/mL). Likewise, Dos Anjos et al., 2021 [150] formulated a cocoa-based ice tea with different concentrations of CBS powder (20,30 and 40 g/L water). The final ice tea that employed 30 g of CBS per liter received the highest organoleptic acceptance, and it was able to retain a maximum of 85% of phenolic compounds from CBS. Moreover, Bernaert and Rysscher (2016) [151] patented the preparation of beverages with alkalized CBS with a high content of dietary fibres.

Despite the great potential of CBS as an additive and/or ingredient in food production, it is necessary to ensure that some of the unacceptable compounds that may be present are at acceptable levels. For example, it has been reported that mycotoxins produced by fungi species such as *Aspergillus*, *Penicillum*, *Absidia* and *Eurotium* concentrate in cocoa shell. Ochratoxin A, aflatoxins B1, B2, G1 and G2 are some examples of the mycotoxins found in CBS [19,152,153]. In addition, genotoxic carcinogens, such as polycyclic aromatic hydrocarbons (PAHs), have also been detected in CBS as a result of an inappropriate drying method or the use of too-low temperatures during the roasting of the beans [154]. In addition, CBS could be contaminated with heavy metals, such as nickel (Ni), chromium (Cr) or cadmium (Cd), from the uncontrolled use of fertilizers, insecticides and pesticides is one of the main reasons for the large contamination of cocoa crops. In fact, the high adsorption capacity of CBS can promote the retention of these harmful compounds [155,156].

### 3.7. Animal Feed

The potential of CBS as an ingredient for animal feed has always been studied in order to reduce the environmental impacts derived from waste disposal, diet costs at the farm level and food–feed competition [56,157]. For example, Emiola et al., 2011 [158] evaluated the effect of CBS feeding on laying hens. The results showed that diets with 15, 20, 25 and 30% CBS resulted in a reduction in egg production compared to control diets and, in addition, spleen, kidney and ovary weights decreased as the percentage of CBS in diets increased. The weights of several internal organs as a percentage of final body weight can be useful as an indicator of the animals’ health [159]. Additionally, Olubaniwa et al., 2006 [160] evaluated the use of CBS boiled during different times (15, 20, 45 and min) to replace maize in 32-weeks old laying hens’ diets. Among the diets evaluated, only the 15-min boiled CBS at 20% maize replacement showed similar egg production and feed conversion as the control diet. CBS was also assessed as feed in monogastric herbivores. Ayinde et al., 2010 [161] analyzed the economic aspects of including CBS as a supplement in rabbit diets. Data showed that the addition of 0.1–0.2 g of CBS per each g of diet improves rabbit growth performance.

Regarding ruminants, Hikmah et al., 2020 [162] studied the effect of diets supplemented with different percentages of CBS (0, 3, 6 and 9%) on cattle bulls, and they concluded that diets with 6 and 9% of CBS showed an increase in liver and kidney compared between 0 and 3%. Renna et al., 2022 [157] investigated the appropriateness of CBS as a feed ingredient in dairy goat diets and evaluated the possible effect on milk yield and composition. These authors reported that milk from goats with CBS-supplemented diets showed a reduction in urea levels and higher concentration of total branched-chain fatty acids compared to those groups without CBS diets. Similarly, Campione et al., 2021 [163], using CBS as a partial substitute for cereal grains in the diet of dairy sheep, observed a decrease in milk urea levels of ewe groups with CBS included in the diet, probably due to the phenolic content of this by-product.

## 4. Conclusions

Due to the economic and environmental issues arising from the accumulation of organic wastes, there is a growing interest in the development of valorization alternatives for these residues. CBS, one of the main by-products, usually managed as waste, which is derived from the chocolate industry, can be used as a raw material in the production of different bioactive compounds that show great interest from a biotechnological point of view due to their potential applications in the food, cosmetic or pharmaceutical sectors. Different extraction techniques (such as extraction with organic solvents), paying special attention to sustainable and environmentally friendly techniques, have been investigated with the aim of maximizing the number of polyphenols obtained from CBS. The obtained extracts have been reported to present beneficial effects on human health since they contain flavanols, particularly catechin and epicatechin, which have antioxidant, antihypertensive, antiatherogenic, anti-inflammatory, hypoglycemic and hypocholesterolemic properties. Additionally, this by-product has been employed in several applications, such as being used as adsorbents to remove emerging contaminants from wastewater, a substrate for biopolymer production, corrosion inhibitors or food ingredients. However, it is worth mentioning that CBS may contain some undesirable pollutants, such as heavy metals or mycotoxins, which could complicate the incorporation of this by-product into food formulations.

According to the literature reviewed in this work regarding CBS valorization, future research should be focused on three approaches: (i) studying the characteristics and properties of the compounds of interest obtained in-depth, (ii) widening the potential applications of this by-product in different fields and (iii) implementing “green” practices that are viable from an economic and technological point of view at an industrial scale to obtain value-added products.

## Figures and Tables

**Figure 1 antioxidants-12-01028-f001:**
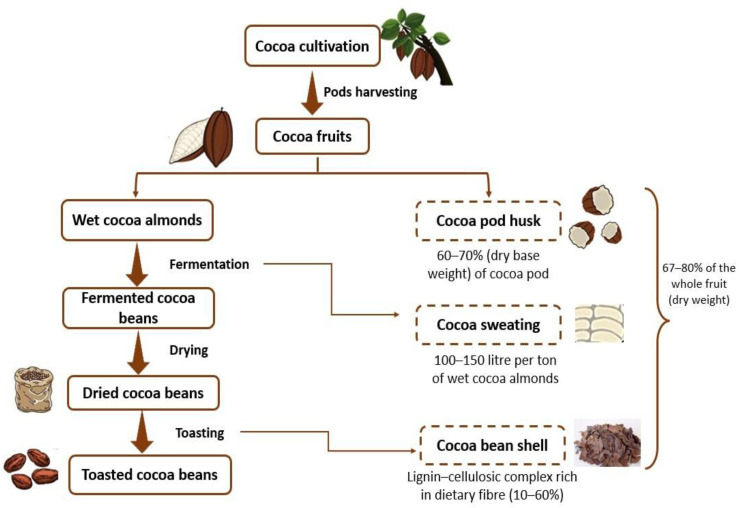
Cocoa processing and derived wastes/by-products. Adapted from [11].

**Figure 2 antioxidants-12-01028-f002:**
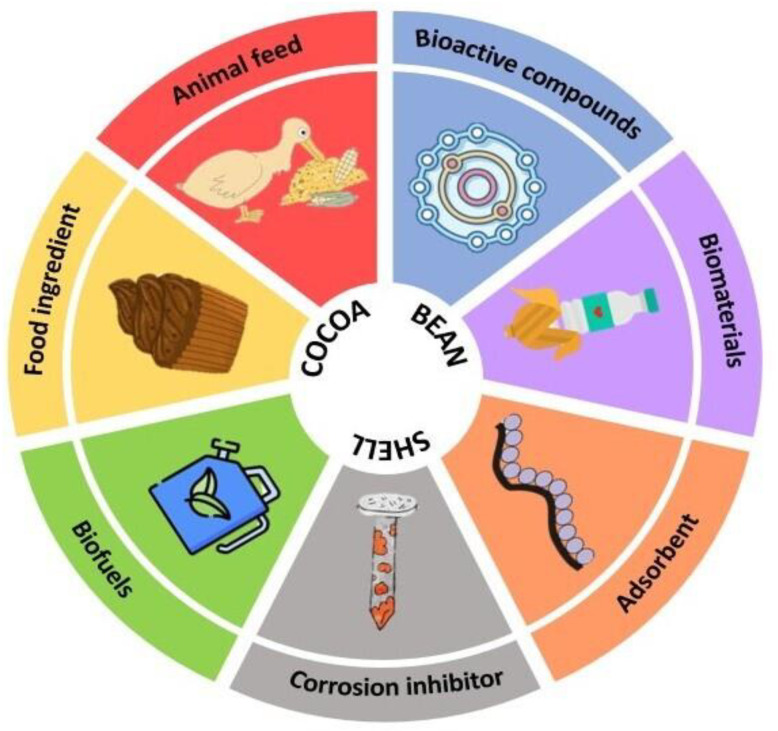
Overview of CBS alternatives of valorization.

**Table 1 antioxidants-12-01028-t001:** Chemical and nutritional composition of CBS, all data, except moisture, expressed on a dry weight basis.

	Content	References
Moisture	4–13.1 g/100 g	[25,27,28,29]
Ash	6.0–9.1 g/100 g	[27,28,30]
Carbohydrates	13.2–70.3 g/100 g	[25,29,31,32,33]
Proteins	18.2–27.4 g/100 g	[27,29,33,34,35]
Lipids	2.3–6.5 g/100 g	[25,27,36,37]
Dietary fibres	13.8–65.6 g/100 g	[25,27,29]
Total phenolic content	22–100 mg GAE/g	[28,30,38,39,40]
Total flavonoid content	7.5–21.8 mg RU/g1.6–43.9 mg CE/g	[39,40,41,42,43]
Total tannin content	2.3–25.3 mg CE/g	[39,40]
Flavanols		
Catechin	0.8–5.7 mg/g	[25,44,45,46]
Epicatechin	0.6–30 mg/g	[25,44,45,46,47]
Procyanidin B1	0.5–0.8 mg/g	[48]
Procyanidin B2	0.2–1.4 mg/g	[48,49]
Methylxanthines		
Theobromine	0.6–13.5 mg/g	[25,30,49,50]
Caffeine	0.1–1.1 mg/g	[30,51]
Theophylline	0.1–0.3 mg/g	[51]

GAE: gallic acid equivalent; RU: rutin equivalent; CE: catechin equivalent.

**Table 2 antioxidants-12-01028-t002:** Bioactive compounds obtained from CBS using different extraction methods, all data, expressed on a dry weight basis.

	Extraction Method	Value	References
Antioxidant activity	Ethanol, methanol–acetone, water, UAE, MAE	2.5–218 µM TE/g CBS	[28,30,39,48]
Total phenolic content	Ethanol, methanol–acetone, water, methanol, acetone, UAE, SWE, PEF, MAE	5.8–154.4 mg GAE/g CBS	[30,32,40,61,63,79,80]
Total flavonoid content	Ethanol, acetone, methanol, PEF	1.6–43.9 mg CE/g CBS	[39,40]
Total tannin content	Ethanol, acetone, methanol, PEF	0.8–25.3 mg CE/g CBS	[39,40]
Total dietary fibre	Ethanol, acetone, methanol	51.8–56.7 g/100 g CBS	[32]
Soluble dietary fibre	Ethanol, acetone, methanol	14.5–16.2 g/100 g CBS	[32]
Insoluble dietary fibre	Ethanol, acetone, methanol	35.6–42.1 g/100 g CBS	[32]
Theobromine	Ethanol, methanol, water, UAE, SWE, PEF, MAE	1.3–11.6 mg/g CBS	[28,30,40,42,79,80,83]
Caffeine	UAE, SWE, PEF, MAE	0.1–4.2 mg/g CBS	[30,40,42,50]
Theophylline	SWE	Traces-0.2 mg/g CBS	[42]
Catechin	Ethanol, water, methanol, PHWE	0.2–6.1 mg/g CBS	[49,61,83,85]
Epicatechin	Ethanol, methanol, water, PEF, MAE, PHWE	0.3–17.7 mg/g CBS	[28,40,48,61,83]
Procyanidin B1	Ethanol, water	0.5–0.8 mg/g CBS	[48]
Procyanidin B2	Ethanol, water	0.2–0.9 mg/g CBS	[48]
Protocatechuic acid	Ethanol, water, MAE	0.9–2.1 mg/g CBS	[28,48]
Caffeic acid	Ethanol, water, methanol, MAE	Traces-0.9 mg/g CBS	[28,61]
Ferulic acid	MAE	0.3–0.5 mg/g CBS	[28]

MAE: Microwave-assisted extraction; PEF: Pulsed-electric-field-assisted extraction; PHWE: Pressurized hot water extraction; SWE: Subcritical water extraction; UAE: Ultrasound-assisted extraction.

## Data Availability

Not applicable.

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
