# Peer review of "Cocoa Bean Shell: A By-Product with High Potential for Nutritional and Biotechnological Applications"

_antioxidants, 2023, doi:10.3390/antiox12051028_

Round 1

Reviewer 1 Report

Page 1 line 36-37: This sentence ”In addition, the cocoa industry generates large amounts of residues, namely, almost 20 tons of waste are produced for each ton of dry cocoa bean obtained” is out of sense. Please correct.

Table 1. Does CBS contain antinutrients such as phytic acid and its lower inositol phosphate forms (IP-% up to IP-1), trypsin inhibitors and more?

Page 12 lines 480-482: Is CBS gluten free and may be applied in gluten free formulations, for example in gluten free bread?

Is any human nutritional study with CBS as a food ingredient were performed recently?

Reviewer 2 Report

The manuscript “Cocoa bean shell: a by-product with high potential for nutritional and biotechnological applications” is a comprehensive and well-written review regarding the beneficial effects of cocoa bean shell, which is found in high quantities after the production of cocoa.

A native English speaker should revise the manuscript as it contains several long and hardly understandable sentences and grammatical errors.

In comparison with several recent review papers (10.1080/10408398.2022.2065659) what does this review paper add to the current knowledge? Please highlight these aspects in more detail.

The authors could also mention the prebiotic effect of this by-product: https://doi.org/10.1080/10408398.2022.2065659

In some titles, sub-titles CBS is found or in abbreviated or under whole format. Please revise and make the article more consistent

line 31 – please put the scientific name in italics, “Theobroma cacao” Revise the entire manuscript

line 65 – if cocoa bean shell has already been abbreviated at lines 47-48, then the abbreviated form should be used afterwards in the manuscript – revise the whole manuscript (i.e. line 70, 216) – or remove the abbreviation

line 79 no > not

line 80 – research > researches

lines 161 – 163 should be rephrased, i.e. “Recently, CBS valorization has been investigated to find novel applications for this by-product, including its use in food formulation, obtention of biofuels, extraction of bioactive compounds and employing it as an adsorbent.”

figure 2 – please provide a higher quality figure, and also this figure could be made more complex

line 184 – second > secondary

line 240 was > were

line 245 – please insert a space between the number and degree sign. Revise the manuscript (the same as line 255)

lines 248 – 252 – rephrase this sentence as it is hard to follow, or separate it in at least two sentences, the same at lines 259-261 (too many and)

lines 306 – 310 – could be rephrased for better understanding: “Results showed that all extracts exhibited antimicrobial power against all strains tested, with the acetone extract presenting the highest inhibition effect. The aqueous extract exhibited a lower antimicrobial effect, whereas ethanolic and methanolic extracts achieved similar inhibition values.”

lines 357 – 359 need a reference – i.e. https://doi.org/10.3390/antiox11091729

lines 366 – 367 – “In recent years, CBS has been investigated for food packaging applications due to its flexural and tensile mechanical properties, reduced density, and low cost.” – is much clearer

lines 494 – 495 needs reference – i.e. https://doi.org/10.3390/foods120510-52

The animal feed section could be further discussed, and the sentence 586-588 – should be better described in the manuscript.

Future perspectives and recommendations based on the current review paper?

After some major revisions and improvements, the manuscript can be considered for publication.

Reviewer 3 Report

This is a very interesting review manuscript addressing the revalorization of the cocoa ben shell (CBS) as a useful by-product of different applications. The topic is very interesting and, as commented by the authors, within the Sustainability Development Goals established by the Agenda 2030, addressing food chain sustainability and efficiency. This is very important for this kind of food product where close to 90% of the total cocoa fruit is discarded as residue.

The nutritional composition of CBS was described in deep based on literature data, and compared with other matrices reported in the literature. CBS valorization is correctly addressed by describing the different applications (food formulation, biofuels, extraction of bioactive substances, adsorbent applications, corrosion inhibitors, food ingredients, and animal feed…). The authors correctly described and discuss the studies reported in the literature. For example, in the case of bioactive substances, different aspects such as health-beneficial properties, contents, and extraction methodologies, are correctly described, for each group of bioactive substances.  

In my opinion, this review manuscript is very well written and provides very useful information for the scientific community dealing specifically with CBS by-products, but also in general for the agri-food waste valorization, a key aspect within a circular economy framework. In my opinion, the review manuscript can be accepted for publication in Antioxidants in its present form. 

Round 2

Reviewer 2 Report

The authors implemented all the required suggestions, and afterwards, it can be accepted for publication.

As a minor detail in line 47, the abbreviation CBS should be described in full as the abstract is considered a separate text.